# Long Non-Coding RNA *MIR31HG* Promotes the Transforming Growth Factor β-Induced Epithelial-Mesenchymal Transition in Pancreatic Ductal Adenocarcinoma Cells

**DOI:** 10.3390/ijms23126559

**Published:** 2022-06-12

**Authors:** Ching-Chung Ko, Yao-Yu Hsieh, Pei-Ming Yang

**Affiliations:** 1Department of Medical Imaging, Chi Mei Medical Center, Tainan 71004, Taiwan; kocc0729@gmail.com; 2Department of Health and Nutrition, Chia Nan University of Pharmacy and Science, Tainan 71710, Taiwan; 3Institute of Biomedical Sciences, National Sun Yat-Sen University, Kaohsiung 80424, Taiwan; 4Division of Hematology and Oncology, Taipei Medical University Shuang Ho Hospital, New Taipei City 23561, Taiwan; alecto39@gmail.com; 5Division of Hematology and Oncology, Department of Internal Medicine, School of Medicine, College of Medicine, Taipei Medical University, Taipei 11031, Taiwan; 6Graduate Institute of Cancer Biology and Drug Discovery, College of Medical Science and Technology, Taipei Medical University, Taipei 11031, Taiwan; 7PhD Program for Cancer Molecular Biology and Drug Discovery, College of Medical Science and Technology, Taipei Medical University and Academia Sinica, Taipei 11031, Taiwan; 8TMU Research Center of Cancer Translational Medicine, Taipei Medical University, Taipei 11031, Taiwan; 9Cancer Center, Wan Fang Hospital, Taipei Medical University, Taipei 11696, Taiwan; 10TMU and Affiliated Hospitals Pancreatic Cancer Groups, Taipei Medical University, Taipei 11031, Taiwan

**Keywords:** epithelial-to-mesenchymal transition, long non-coding RNA, metastasis, pancreatic ductal adenocarcinoma, transforming growth factor β

## Abstract

The epithelial-to-mesenchymal transition (EMT) describes a biological process in which polarized epithelial cells are converted into highly motile mesenchymal cells. It promotes cancer cell dissemination, allowing them to form distal metastases, and also involves drug resistance in metastatic cancers. Transforming growth factor β (TGFβ) is a multifunctional cytokine that plays essential roles in development and carcinogenesis. It is a major inducer of the EMT. The MIR31 host gene (*MIR31HG*) is a newly identified long non-coding (lnc)RNA that exhibits ambiguous roles in cancer. In this study, a cancer genomics analysis predicted that *MIR31HG* overexpression was positively correlated with poorer disease-free survival of pancreatic ductal adenocarcinoma (PDAC) patients, which was associated with upregulation of genes related to TGFβ signaling and the EMT. In vitro evidence demonstrated that TGFβ induced *MIR31HG* expression in PDAC cells, and knockdown of *MIR31HG* expression reversed TGFβ-induced EMT phenotypes and cancer cell migration. Therefore, *MIR31HG* has an oncogenic role in PDAC by promoting the EMT.

## 1. Introduction

In 2020, pancreatic cancer became the seventh leading cause of cancer mortality worldwide, and there are nearly as many pancreatic cancer patient deaths (466,000) as there are new cases (496,000) due to its poor prognosis [1]. Pancreatic ductal adenocarcinoma (PDAC) accounts for more than 85% of patients, and it is the most prevalent and aggressive pancreatic cancer type [2]. PDAC has a very low 5-year survival rate (of <8%) because of late diagnoses, high rates of metastasis, and drug resistance [3,4]. The recurrence rate remains significant even in patients with early-stage disease [5]. The majority of PDAC patients (80%~85%) are not suitable for surgical resection, which is the only curative therapeutic option. Furthermore, approximately half of PDAC patients have metastases when first diagnosed, also negating curative resection [4]. Gemcitabine-based chemotherapy is indispensable for treating unresectable pancreatic cancer [5]. However, current regimens are not satisfactory because PDAC is one of the most chemoresistant cancer types [6]. Therefore, obtaining a better understanding of pancreatic cancer biology will help in developing novel therapeutic strategies.

Dysregulation of long non-coding (lnc)RNAs is frequently found in cancers, which reveals new targets for interventions [7,8]. The MIR31 host gene (*MIR31HG*) is an lncRNA located on human chromosome 9 (9p21.3). Although *MIR31HG* is the host gene of microRNA (miR)-31 and their expressions are positively correlated in some cancers, knockdown of *MIR31HG* does not change the level of miR-31 [9,10], suggesting that *MIR31HG* may function in cancers independent of miR-31. *MIR31HG* is frequently upregulated in various cancer types, including PDAC, and serves as an oncogene and a poor prognostic factor [9,10,11,12,13,14,15,16,17,18,19,20]. Previously identified *MIR31HG* targets include hypoxia-inducible factor (HIF)-1α, p21, miR-193b, miR-214, miR-361-3p, and miR-761, and these are associated with tumor growth, metastasis, and chemoresistance [10,11,12,14,17,20,21]. As a co-activator of HIF-1α, *MIR31HG* is also named long non-coding HIF-1α co-activating RNA (*lncHIFCAR*) [14]. Controversially, *MIR31HG* is downregulated in bladder cancer, esophageal squamous cell carcinoma, and hepatocellular carcinoma [22,23,24]. Therefore, the role of *MIR31HG* in cancers might be cancer type-specific.

The epithelial-to-mesenchymal transition (EMT) describes a biological process in which polarized epithelial cells are converted into highly motile mesenchymal cells. Loss of cell–cell adhesion and related markers such as E-cadherin (CDH1), upregulation of mesenchymal markers such as vimentin (VIM) and N-cadherin (CDH2), and acquisition of a motile capacity and a fibroblast-like phenotype, are major hallmarks of the EMT [25,26]. The EMT plays an important role during embryogenesis, such as facilitating the generation of new tissues and organs. It also contributes to the pathogenesis of tissue fibrosis, tumor progression, and metastasis [27,28,29]. One of the most distinguishing and significant characteristics of pancreatic cancer is the EMT. It occurs even in pancreatic intraepithelial neoplasia (PanIN), the histological precursor to invasive PDAC, and leads to early dissemination, drug resistance, and a poor prognosis [30]. Although the precise role of the EMT in the pancreatic cancer cells’ biological behaviors and its implications for clinical therapy remain controversial, a therapeutic strategy of combining EMT inhibition with chemotherapy is worth considering [31].

In this study, cancer genomics data mining revealed that *MIR31HG* overexpression was positively associated with poorer disease-free survival and a transforming growth factor β (TGFβ)-induced EMT gene signature in PDAC patients. In vitro experiments demonstrated that TGFβ induced *MIR31HG* expression and promoted the TGFβ-induced EMT in PDAC cells. Our results support the oncogenic role of *MIR31HG* in PDAC.

## 2. Results

### 2.1. Upregulation of MIR31HG Is Associated with Disease-Free Survival in PDAC Patients

Previously, only one study investigated the role of *MIR31HG* in pancreatic cancer [10], which found that *MIR31HG* is overexpressed in PDAC tissues and cell lines. Downregulation of *MIR31HG* inhibits in vitro and in vivo PDAC cell growth by repressing cell cycle progression and inducing apoptosis. In addition, they identified that a tumor-suppressive miR-193b directly targets *MIR31HG*, and *MIR31HG* also competes for miR-193b binding to its messenger (m)RNA targets [10]. To obtain a greater understanding of the role of *MIR31HG* in PDAC, *MIR31HG* expressions in normal and PDAC tissues were obtained from the Gene Expression Profiling Interactive Analysis (GEPIA) database [32]. As shown in Figure 1A, *MIR31HG* was overexpressed in PDAC tumor tissues according to The Cancer Genome Atlas (TCGA) PanCancer Atlas dataset [33]. Because only four normal pancreatic tissues existed in this dataset and an additional 167 normal pancreatic tissues were from Genotype-Tissue Expression (GTEx) data [34], two other pancreatic cancer cohorts (GSE16515 [35] and GSE28735 [36]) were obtained from the Gene Expression Omnibus (GEO) database [37]. Consistently, *MIR31HG* was found to be overexpressed in tumor tissues in these two cohorts (Figure 1B). To investigate the prognostic role of *MIR31HG* in PDAC patients, Kaplan-Meier overall and disease-free survival plots correlated with *MIR31HG* expression were obtained from the GEPIA and PROGgeneV2 databases [38]. As shown in Figure 1C and 1D, *MIR31HG* overexpression was not correlated with overall survival in either TCGA or GSE28735 datasets. However, *MIR31HG* expression was correlated with disease-free survival in the TCGA dataset, suggesting that *MIR31HG* overexpression may be correlated with recurrence and metastasis in PDAC patients. Because the disease-free survival status in the GSE28735 cohort was not available, whether *MIR31HG* overexpression was also correlated with patient’s disease-free survival was unknown.

### 2.2. Upregulation of MIR31HG Is Associated with the EMT Gene Signature in PDAC Patients

To investigate molecular alterations correlated with *MIR31HG* overexpression in PDAC, microarray gene expression profiles in GSE16515 and GSE28735 were analyzed using Gene Set Enrichment Analysis (GSEA) software [39,40] for enrichment of cancer hallmarks [41]. We found that seven cancer hallmarks, including TGFβ signaling, were commonly enriched in the two PDAC cohorts (Figure 2A). TGFβ signaling has both protumorigenic and tumor-suppressive roles in PDAC, which depends on the tumor stage [42,43]. TGFβ suppresses tumors in the early stages of carcinogenesis, because it serves as an antimitogen that stops cell-cycle progression during the G_1_ phase [44]. During pancreatic carcinogenesis, changes in TGFβ signaling components are prevalent. For example, mutations in the *SMAD4* and TGFβ type II receptor (*TGFBRII*) genes may make cancer cells resistant to TGFβ’s antimitogenic activity [42,43]. Thus, TGFβ promotes cancer invasion, angiogenesis, and metastasis by inducing the EMT in late-stage carcinogenesis [42,43]. Although the EMT cancer hallmark was not commonly enriched in these two pancreatic cancer cohorts, we found that the EMT tended to be enriched in *MIR31HG*-high expressing pancreatic cancer patients (Figure 2B). Therefore, *MIR31HG* overexpression was correlated with the EMT gene signature in PDAC. Consistent with our study, colorectal cancer patients with higher *MIR31HG* expression were characterized by elevated TGFβ and EMT gene expressions [15].

### 2.3. Upregulation of MIR31HG Is Associated with the TGFβ-Induced EMT in PANC-1 Cells

lncRNAs, such as *MALAT1*, *HOTAIR*, *H19*, *LncRNA-ATB*, and *LincRNA-ROR*, are involved in the EMT through cross-talk with their master regulators [45,46]. However, the role of *MIR31HG* in the EMT is still unclear. To investigate the role of *MIR31HG* in the TGFβ-induced EMT, a microarray dataset (GSE23952 [47]) from TGFβ-treated PANC-1 cells was obtained from the GEO database. Relative mRNA expression levels of *MIR31HG* and EMT markers were visualized as a heat map (Figure 3A). We found that *MIR31HG* expression was higher during the TGFβ-induced EMT in PANC-1 cells, along with induction of mesenchymal markers (zinc finger E-box binding homeobox 1 (*ZEB1*), *ZEB2*, Snail (*SNAI1*), Slug (*SNAI2*), twist family bHLH transcription factor 1 (*TWIST1*), *VIM*, *CDH2*, fibronectin 1 (*FN1*), and collagen type I alpha 1 chain (*COL1A1*)) and reduction of epithelial markers (*CDH1*, P-cadherin (*CDH3*), plakoglobin (*JUP*), desmoplakin (*DSP*), plakophilin 2 (*PKP2*), claudin 3 (*CLDN3*), and *CLDN4*) (Figure 3A). To confirm this observation, PANC-1 cells were challenged with TGFβ, and selected EMT markers were detected by real-time quantitative polymerase chain reaction (qPCR) and Western blot analyses. As shown in Figure 3B, TGFβ indeed induced *MIR31HG* in PANC-1 cells. Induction of the EMT was ascertained by the upregulation of VIM, CDH2, COL1A1, SNAI1, and SNAI2, and the downregulation of CDH1, CLDN4, and JUP at the mRNA (Figure 3B) and/or protein levels (Figure 3C). Cell morphological changes further confirmed this phenomenon. Untreated cells showed a pebble-like shape and cell–cell adhesion. After treatment with TGFβ for 72 h, cells had converted to a spindle-shaped, fibroblast-like morphology (Figure 3D). Therefore, *MIR31HG* was induced during the TGFβ-induced EMT.

### 2.4. MIR31HG Enhances the TGFβ-Induced EMT in PANC-1 Cells

To investigate whether *MIR31HG* participates in the TGFβ-induced EMT, PANC-1 cells were transfected with *MIR31HG* small interfering (si)RNA and then challenged with TGFβ. Cell morphological observations showed that cells had converted to a spindle-shaped, fibroblast-like morphology by 48 h TGFβ treatment, which was prevented by *MIR31HG* knockdown (Figure 4A). Real-time qPCR and Western blot analyses showed that *MIR31HG* knockdown suppressed TGFβ-induced downregulation of CDH1 and CLDN4 and upregulation of COL1A1 at both the mRNA and protein levels (Figure 4B,C). On the other hand, *MIR31HG* overexpression slightly enhanced 24 h TGFβ treatment-induced cell morphological changes and upregulation of COL1A1 and VIM at the mRNA or protein level (Figure 4D–F). However, *MIR31HG* knockdown or overexpression alone was insufficient to induce significant changes in cell morphology and EMT marker expressions (Figure 4), suggesting that *MIR31HG* participates in the EMT when TGFβ signaling is activated. Acquisition of a motile capacity is an important feature of the EMT [48]. Thus, the migrating activity of PANC-1 cells was measured by a wound-healing assay. As shown in Figure 5, TGFβ increased the cell-migrating activity, which was rescued by silencing the *MIR31HG* expression. Consistently, *MIR31HG* knockdown alone did not affect cell-migratory activity.

## 3. Discussion

Depending on the cancer type, *MIR31HG* can be either oncogenic or tumor-suppressive [9,10,11,12,13,14,15,16,17,18,19,20,22,23,24]. For example, *MIR31HG* inhibits cell proliferation and metastasis in hepatocellular carcinoma [24]. In contrast, *MIR31HG* acts as an HIF-1α co-activator and drives oral cancer progression [14]. The oncogenic property of *MIR31HG* was also identified in PDAC, in which *MIR31HG* promotes cancer cell proliferation and invasion [10]. Our results also support the oncogenic role of *MIR31HG* in PDAC by enhancing the TGFβ-induced EMT.

Cancer cells employ the EMT to gain invasive ability [48]. It was shown that *MIR31HG* knockdown inhibited the invasion of AsPC-1 and PANC-1 pancreatic cancer cells [10]. Whether *MIR31HG* knockdown also inhibits TGFβ-induced cell invasion warrants further investigation. The above study together with ours also implies that the effects of *MIR31HG* on cancer cell migration and invasion are uncoupled as reported earlier in other cancer cell types [49,50]. In addition, previous studies in non-small cell lung cancer and osteosarcoma cells also identified a promoting role of *MIR31HG* in the EMT [51,52]. However, they found that *MIR31HG* knockdown was sufficient to upregulate mesenchymal markers and downregulate epithelial markers [52], suggesting that a cancer type-specific role of *MIR31HG* may exist.

Our results indicated that *MIR31HG* alone did not seem to impact the EMT phenotype. However, loss-of-function and gain-of-function experiments indicated that *COL1A1* gene expression was regulated by *MIR31HG* among the EMT markers tested in this study, suggesting that *COL1A1* may be a potential target of *MIR31HG*. Interestingly, *COL1A1* expression can be suppressed by miR-193 family members [53,54]. Because *MIR31HG* acts as an endogenous sponge of miR-193b in PDAC [10], we hypothesized that *MIR31HG* may also compete for miR-193 binding to *COL1A1* mRNA.

In addition to its role in cancer, *MIR31HG* also plays a role in adipocyte differentiation (adipogenesis) [55]. Overexpression of *MIR31HG* reduces adipocyte differentiation in vitro and in vivo. In contrast, knockdown of *MIR31HG* inhibits the expression of an adipogenic-related gene, fatty acid binding protein 4 (*FABP4*), via histone modification of its gene promoter, leading to suppression of adipocyte differentiation [55]. Interestingly, it has been reported that EMT-derived breast cancer cells, but not epithelial cancer cells, can be trans-differentiated into post-mitotic and functional adipocytes, leading to the repression of primary tumor metastasis [56]. It would be interesting to explore whether such phenomena also exist in PDAC and the involving role of MIR31HG in cancer cell-adipocyte trans-differentiation in the future.

In conclusion, we found that *MIR31HG* overexpression in PDAC was positively associated with patients’ disease-free survival, but not overall survival. In addition, *MIR31HG* overexpression was correlated with upregulation of TGFβ signaling and the EMT. In vitro experiments confirmed the induction of *MIR31HG* by TGFβ treatment in PDAC cells. Knockdown of MIR31HG expression reversed the TGFβ-induced EMT. However, this study has several potential biases and limitations. First, the expression of *MIR31HG* and its clinical impact in PDAC patients were only analyzed using TCGA, GSE16515, and GSE28735 datasets. Further validation using clinical samples is still needed. Second, only one cell line was used in the in vitro experiments. Cell type variations may exist. In addition, animal experiments are needed to validate the in vitro observations. Third, the mechanisms of how TGFβ upregulates *MIR31HG* expression and how *MIR31HG* promotes the TGFβ-induced EMT have not yet been elucidated. Therefore, more investigations are required for a full understanding of the oncogenic role of *MIR31HG* in PDAC and a better validation of our conclusions.

## 4. Materials and Methods

### 4.1. Cell Culture and Treatment

A human pancreatic cancer cell line (PANC-1) from the Bioresource Collection and Research Center (BCRC; Hsinchu, Taiwan) was kindly provided by Prof. Hsin-Yi Chen (Taipei Medical University, Taipei, Taiwan). Cells were cultured in Dulbecco’s modified Eagle medium (DMEM) containing 10% fetal bovine serum (FBS), 2 mM L-glutamine, 1 mM sodium pyruvate, and 1% antibiotic-antimycotic, and were grown at 37 °C in a humidified CO_2_ (5%) incubator. For TGFβ treatment, cells were first serum-starved for 24 h and then treated with TGFβ in serum-free medium. The recombinant human TGFβ1 (#100-21) was purchased from PeproTech (Rocky Hill, NJ, USA). Cellular morphological changes were photographed at 20× magnification under an inverted microscope (IX71, Olympus, Tokyo, Japan).

### 4.2. Real-Time Quantitative Polymerase Chain Reaction (qPCR)

Total RNA was purified with a GENEzol TriRNA Pure Kit (#GZX100; Geneaid, New Taipei City, Taiwan). First-strand complementary (c)DNA was synthesized using an iScript cDNA Synthesis Kit (#1708891; Bio-Rad Laboratories, Hercules, CA, USA). PCR amplification with gene-specific primers (Table 1) was performed using the IQ2 SYBR Green Fast qPCR System Master Mix (#DBU-006; Bio-Genesis Technologies, Taipei, Taiwan) on a LightCycler 96 System (Roche, Indianapolis, IN, USA).

### 4.3. Western Blot Analysis

Whole-cell lysates were extracted by lysing cells in the radioimmunoprecipitation assay (RIPA) buffer containing 1× protease and phosphatase inhibitor cocktails. After separation by sodium dodecylsulfate-polyacrylamide gel electrophoresis (SDS-PAGE), proteins were transferred to nitrocellulose membranes. Membranes were blocked with 5% skimmed milk in TBST buffer (20 mM Tris-base, 150 mM NaCl, and 0.05% Tween-20), and then blotted with a specific primary antibody and a corresponding horseradish peroxidase (HRP)-conjugated secondary antibody. Protein bands were developed with the Western Lightning Plus ECL detecting reagent (#NEL105001EA; PerkinElmer, Waltham, MA, USA) and detected using GE Amersham Imager 600 (GE Healthcare Life Sciences, Marlborough, MA, USA). E-cadherin (CDH1; #3195), N-cadherin (CDH2; #13116), and plakoglobin (JUP; #2309) antibodies were purchased from Cell Signaling Technology (Beverly, MA, USA). Claudin 4 (CLDN4; #GTX108483), collagen type I alpha 1 chain (COL1A1; #GTX112731), vimentin (VIM; #GTX100619), and GAPDH (#GTX100118) antibodies were purchased from GeneTex (Hsinchu, Taiwan). Uncropped images for each blot are shown in Appendix A.

### 4.4. Transfection

For the *MIR31HG*-knockdown analysis, small interfering (si)RNAs were transiently transfected into cells using the Lipofectamine RNAiMAX reagent (#13778150; ThermoFisher Scientific, Waltham, MA, USA). Lincode *MIR31HG* siRNA (SMARTpool; #R-187931-00-0005) and its non-targeting siRNA (#D-001320-10-05) were purchased from Horizon Discovery (Cambridge, UK). For the *MIR31HG*-overexpression analysis, pSL-MS2 (pMS2) and pSL-MS2-MIR31HG plasmids [14], kindly provided by Prof. Jing-Wen Shih (Taipei Medical University), were transfected into cells using the Lipofectamine 3000 reagent (#L3000015; ThermoFisher Scientific). Twenty-four hours after transfection, cells were used for further experiments.

### 4.5. Wound-Healing Assay

Cells (10^6^) were grown in six-well plates for 24 h to reach a monolayer at more than 90% confluence. A wound in each well was created with a pipette tip. Floating cells were washed away with prewarmed phosphate-buffered saline (PBS), and then cells were treated with TGFβ in serum-free medium. Wounds were photographed in the same position at 0 and 48 h. Wound sizes were analyzed by an ImageJ plugin (Wound_healing_size_tool) [57]. The wound closure rate was calculated by the following formula: Wound closure (%) = [(A_0h_ − A_48h_)/A_0h_] × 100%, where A_0h_ is the initial wound size at time zero and A_48h_ is the wound size after 48 h.

### 4.6. Statistical Analysis

Statistical analyses were performed using GraphPad Prism 9 (GraphPad Software, San Diego, CA, USA). Data were initially tested for normality using the Kolmogorov–Smirnov test. Parametric (normally distributed) data were analyzed using an unpaired two-tailed Student′s t-test. Nonparametric (non-normally distributed) data were analyzed with the two-tailed Mann–Whitney test. A *p* value of <0.05 was considered statistically significant.

## Figures and Tables

**Figure 1 ijms-23-06559-f001:**
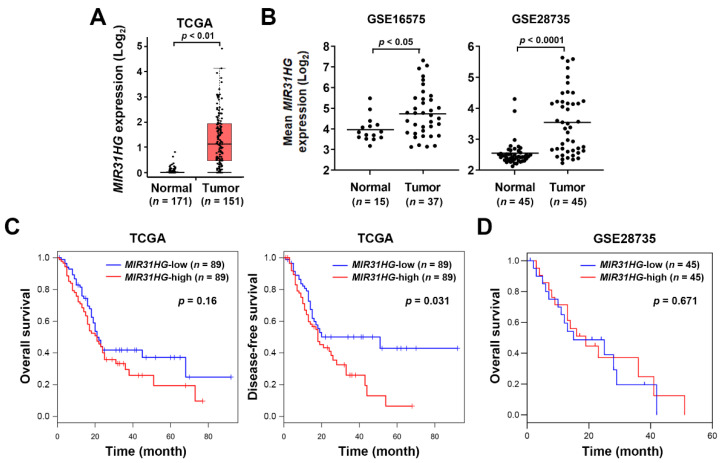
*MIR31HG* overexpression in pancreatic ductal adenocarcinoma (PDAC) patients: (**A**) *MIR31HG* expressions in pancreatic tumor and normal tissues were obtained from the GEPIA database, while 151 pancreatic tumors and four normal tissues were from the TCGA PanCancer Atlas dataset. Another 167 normal pancreatic tissues were from the GTEx database; (**B**) *MIR31HG* expressions in normal and pancreatic cancer tissues were obtained from two pancreatic cancer cohorts (GSE16515 and GSE28735) in the GEO database; (**C**) Kaplan-Meier overall and disease-free survival plots for cancer patients (TCGA PanCancer Atlas dataset) with high and low *MIR31HG* expression were generated using the GEPIA database; (**D**) Kaplan-Meier overall survival plots for cancer patients (GSE28735) with high and low *MIR31HG* expression were generated using PROGgeneV2 database.

**Figure 2 ijms-23-06559-f002:**
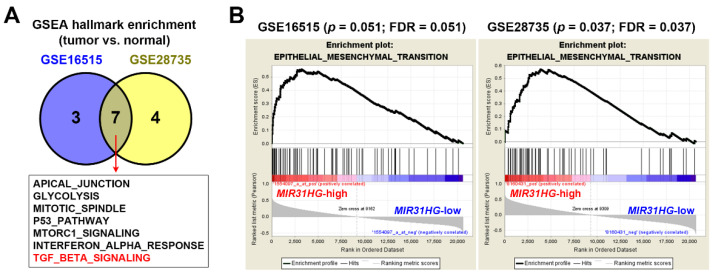
Gene set enrichment analysis (GSEA) revealed the potential role of *MIR31HG* in transforming growth factor β (TGFβ) signaling and the epithelial-to-mesenchymal transition (EMT): (**A**) A GSEA was performed to enrich 50 cancer hallmarks in pancreatic tumor tissues; (**B**) A GSEA was performed to enrich the EMT gene signature in two pancreatic cancer cohorts (GSE16515 and GSE28735). FDR, false-discovery rate.

**Figure 3 ijms-23-06559-f003:**
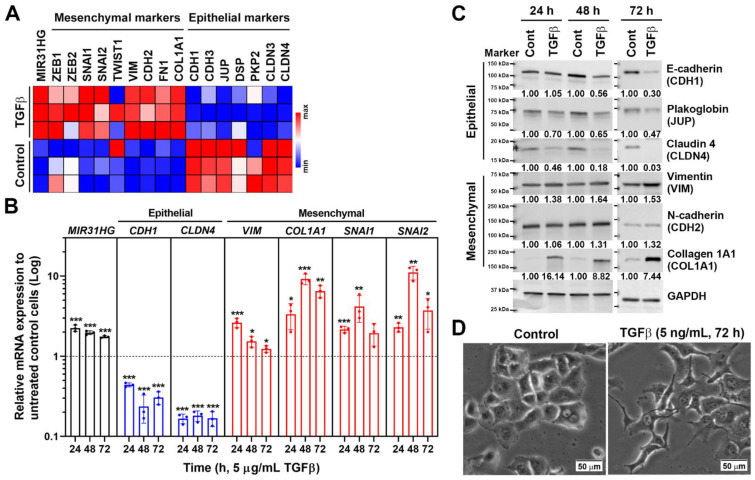
Transforming growth factor β (TGFβ) induced *MIR31HG* and the epithelial-to-mesenchymal transition (EMT) in PANC-1 cells: (**A**) A microarray dataset (GSE23952) for TGFβ (5 ng/mL for 48 h)-treated PANC-1 cells was obtained from the GEO database. A heat map shows the relative expressions of *MIR31HG* and EMT markers; (**B**) PANC-1 cells were treated with 5 ng/mL TGFβ for 24, 48, and 72 h. Total RNAs were examined by a real-time qPCR for *MIR31HG* and EMT marker expressions. Data represent the fold changes of mRNA expression (mean ± standard deviation, SD) for each gene compared to untreated control cells at each time point. * *p* < 0.05, ** *p* < 0.01, and *** *p* < 0.001 indicate statistical significance compared to untreated control cells; (**C**) PANC-1 cells were treated with 5 ng/mL TGFβ for 24, 48, and 72 h. Total protein lysates were examined by Western blotting for EMT marker expressions. Representative images of each protein were obtained from the same or different blots. The band intensity was quantified and the related ratio to each time point was shown; (**D**) PANC-1 cells were treated with 5 ng/mL TGFβ for 72 h. The cell morphology was observed under bright-field microscopy. Scale bar, 50 μm.

**Figure 4 ijms-23-06559-f004:**
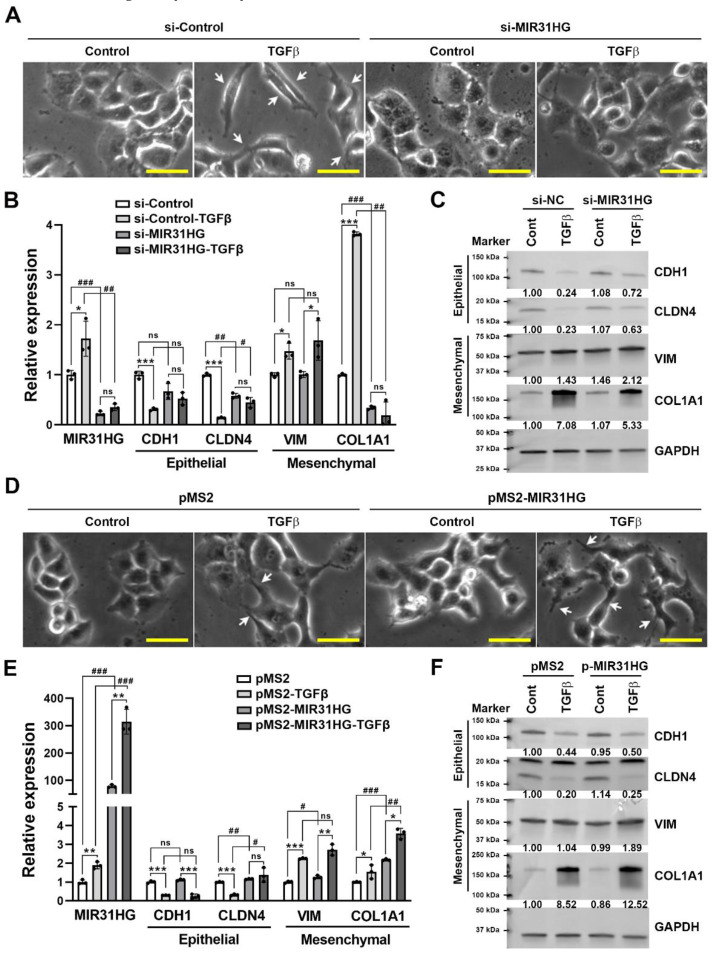
Knockdown of *MIR31HG* reversed transforming growth factor β (TGFβ)-induced epithelial-to-mesenchymal transition (EMT) phenotypes: (**A**) PANC-1 cells were transfected with *MIR31HG* or control siRNA for 24 h and then exposed to 5 ng/mL TGFβ for another 48 h. The cell morphology was observed under bright-field microscopy. The arrows indicate cells with spindle-shaped, fibroblast-like morphology. Scale bar, 25 μm; (**B**) PANC-1 cells were transfected with *MIR31HG* or control siRNA for 24 h and then exposed to 5 ng/mL TGFβ for another 24 h. Total RNAs were examined by a real-time qPCR for *MIR31HG* and EMT marker expressions. Results are shown as the mean ± standard deviation (SD). * *p* < 0.05 and *** *p* < 0.001 or ^#^
*p* < 0.05, ^##^
*p* < 0.01, and ^###^
*p* < 0.001 indicate statistical significance between groups. ns, no statistical significance between indicated groups; (**C**) PANC-1 cells were transfected with *MIR31HG* or control siRNA for 24 h and then exposed to 5 ng/mL TGFβ for another 48 h. Total protein lysates were examined by Western blotting for EMT marker expressions. Representative images of each protein were obtained from the same or different blots. The band intensity was quantified and the related ratio to untreated si-NC cells was shown; (**D**) PANC-1 cells were transfected with a *MIR31HG*-encoding or control plasmid for 24 h and then exposed to 5 ng/mL TGFβ for another 24 h. Cell morphology was observed under bright-field microscopy. The arrows indicate cells with spindle-shaped, fibroblast-like morphology. Scale bar, 25 μm; (**E**) PANC-1 cells were transfected with a *MIR31HG*-encoding or control plasmid for 24 h and then exposed to 5 ng/mL TGFβ for another 24 h. Total RNAs were examined by a real-time qPCR for *MIR31HG* and EMT marker expressions. Results are shown as the mean ± standard deviation (SD). * *p* < 0.05, ** *p* < 0.01, and *** *p* < 0.001 or ^#^
*p* < 0.05, ^##^
*p* < 0.01, and ^###^
*p* < 0.001 indicate statistical significance between groups. ns, no statistical significance between indicated groups; (**F**) PANC-1 cells were transfected with a *MIR31HG*-encoding or control plasmid for 24 h and then exposed to 5 ng/mL TGFβ for another 48 h. Total protein lysates were examined by Western blotting for EMT marker expressions. Representative images of each protein were obtained from the same or different blots. The band intensity was quantified and the related ratio to untreated pMS2 cells was shown.

**Figure 5 ijms-23-06559-f005:**
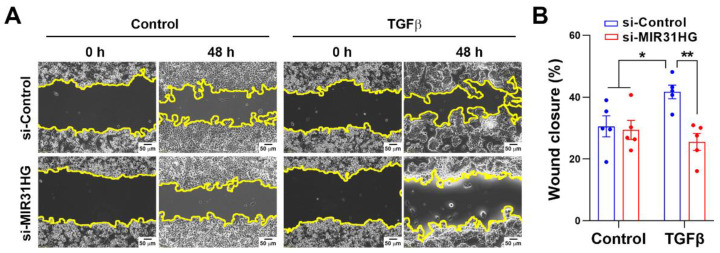
Knockdown of *MIR31HG* reversed transforming growth factor β (TGFβ)-induced cell migration: (**A**) PANC-1 cells were transfected with *MIR31HG* or control siRNA for 24 h and then a wound was created and photographed before (0 h) and after treatment with 5 ng/mL TGFβ for 48 h. Cell migration was examined by the wound closure rate. Scale bar, 50 μm; (**B**) Results in (**A**) were quantified (mean ± standard error of the mean, SEM). * *p* < 0.05 and ** *p* < 0.01 indicate statistical significance between groups.

**Table 1 ijms-23-06559-t001:** Primer pairs used in this study.

Gene	Sequence	Product Length
*MIR31HG*	Forward: 5′-CACCAAGGTGTTCCTGCCTA-3′	147 bp
	Reverse: 5′-CAACCAGGCCAAAAGCATCC-3′	
*CDH1*	Forward: 5′-TACACTGCCCAGGAGCCAGA-3′	103 bp
	Reverse: 5′-TGGCACCAGTGTCCGGATTA-3′	
*CLDN4*	Forward: 5′-CGCATCAGGACTGGCTTTATCTC-3′	187 bp
	Reverse: 5′-CAGCGCGATGCCCATTA-3′	
*VIM*	Forward: 5′-AGTCCACTGAGTACCGGAGAC-3′	98 bp
	Reverse: 5′-CATTTCACGCATCTGGCGTTC-3′	
*COL1A1*	Forward: 5′-CGGAGGAGAGTCAGGAAGG-3′	153 bp
	Reverse: 5′-ACATCAAGACAAGAACGAGGTAG-3′	
*SNAI1*	Forward: 5′-ACCACTATGCCGCGCTCTT-3′	115 bp
	Reverse: 5′-GGTCGTAGGGCTGCTGGAA-3′	
*SNAI2*	Forward: 5′-TGTTGCAGTGAGGGCAAGAA-3′	72 bp
	Reverse: 5′-GACCCTGGTTGCTTCAAGGA-3′	
*β-actin*	Forward: 5′-GTTGCTATCCAGGCTGTGCT-3′	113 bp
	Reverse: 5′-AGGGCATACCCCTCGTAGAT-3′	

## Data Availability

The data supporting this study can be obtained from the public databases or are available on request from the corresponding author.

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
