# Peer review of "Long Non-Coding RNA MIR31HG Promotes the Transforming Growth Factor β-Induced Epithelial-Mesenchymal Transition in Pancreatic Ductal Adenocarcinoma Cells"

_ijms, 2022, doi:10.3390/ijms23126559_

Round 1

Reviewer 1 Report

The current revised submission has extensively improved and the authors have responded adequately to the previous comments. The updated discussion also improved the paper. Hence, there are no further comments and would recommend the editor consider the paper for the journal.

Reviewer 2 Report

The manuscript by Yang and co-workers proposed an oncogenic role of a long noncoding RNA MIR31HG in TGF β induced EMT in Pancreatic Ductal Adenocarcinoma Cells. They performed cancer genomics analysis from different databases and proposed a positive correlation of MIR31HG expression with poor disease-free survival of Pancreatic Ductal Adenocarcinoma Cells patients. By applying real-time qPCR and western blot they demonstrated that RNAi mediated knockdown of MIR31HG reversed TGFβ-induced EMT phenotypes and cancer cell migration. The interpretations of the manuscript are very promising but some of the experimental observations are not matching with their interpretation and they need further analysis.

Specific comments.

1.      This is not clear why the relation between MIR31HG expression and disease-free survival in the GSE28735 pancreatic cancer cohort has not been shown.

2.      In figure 3B- the RT PCR data is shown only after 24 hr while cell morphology has been observed after 72 hr of the treatment. The author should provide the expression of those transcripts also after 48 hr and 72 hr of TGF treatment.

3.      Figure 3C and 4F- the western blot shows no clear difference in CDH2 expression even after 72 hr treatment. This needs proper justification. Quantification of all the western blot images would be helpful to understand the data.

4.      Figure 4A and 4D - The cell morphology is indistinct in TGF-b treatment in MIR32HG RNAi cells. The microscopic imaging setup has not mentioned in the method section. The author should provide a good quality phase contrast image to visualize the phenotype mentioned in the text. They should highlight the cell region showing extensions.  

5.      Figure 4B-C Mention the duration of treatment of RNAi and TGF-B.

Minor Comments

6.      Page 10 line 318- mention the name of the Image J plugin used in the measurement.

7.      Page 2 lines 87-88- mention the related reference  

Round 2

Reviewer 2 Report

None

Author Response

We thank the Reviewer’s effort in improving this manuscript. There is no further comment that we need to respond.